# Privacy Challenges in Conversational AI: Three Use Cases and Prospects for Decentralised Data Governance with Solid

[Paola Di Maio][1]

[1] [Ronin Scholar]

[paola.dimaio@ronin-institute,org]

**Abstract.**

*Conversational AI systems accumulate intimate, longitudinal records of user interactions -- including personal disclosures, intellectual property, health information, and private reflections -- under conditions of structural privacy asymmetry. This paper presents three empirically grounded use cases documenting privacy failures in centralised conversational AI platforms: (1) an assurance gap between verbal confidentiality promises and documented policy, (2) unverifiable data access by AI agents operating through external service connectors, and (3) undocumented scope creep through unannounced screen-sharing capabilities. For each use case, we analyse the privacy failure, map it to relevant regulatory frameworks (GDPR, EU AI Act), and assess how the Solid protocol's decentralised architecture -- user-controlled Pods, granular access control, and linked data standards -- could address the identified gaps. We draw on prior work including SocialGenPod and the W3C Data Privacy Vocabulary (DPV) to contextualise our proposal, while identifying limitations that decentralisation alone cannot resolve. We frame these use cases as a research agenda for the Solid community, proposing extensions including Pod-based conversation storage, standardised audit logging, and machine-readable consent records for AI interactions.*

**Keywords:** Conversational AI, privacy, Solid, decentralised data governance, personal data, GDPR, audit logging, consent, user-controlled data

## 1. Introduction

Conversational AI systems such as ChatGPT, Claude, and Gemini have become integral to professional and personal workflows. Unlike traditional web applications that collect discrete data points, conversational AI accumulates rich, longitudinal records of natural language interaction that may contain personal health disclosures, intellectual property, private reflections, and sensitive professional communications [1]. The conversational interface encourages openness, yet the privacy protections governing the resulting data remain ambiguous and structurally asymmetric.

This paper presents three use cases grounded in empirical experience with a commercial conversational AI platform. Each documents a specific privacy failure mode that arises from the centralised architecture of current systems and the absence of adequate standards for user data governance in conversational AI. The use cases are developed collaboratively within the W3C AI related communities as part of ongoing work on privacy standards for AI systems [2].

We assess how the Solid protocol [3, 4] -- a W3C-standardised architecture for decentralised personal data management -- could address the identified failures. Our analysis builds on prior work demonstrating the feasibility of Solid-based generative AI applications [5] and on the W3C Data Privacy Vocabulary (DPV) [6], which provides machine-readable representations for privacy-relevant metadata compatible with Solid. We frame the use cases not as definitive solutions but as a research agenda proposing specific extensions to the Solid ecosystem for conversational AI data governance.

This paper makes three contributions: (1) three empirically grounded use cases documenting distinct privacy failure modes in centralised conversational AI platforms; (2) a systematic mapping of each failure to Solid architectural mechanisms, with explicit identification of limitations; and (3) a concrete research agenda proposing extensions to the Solid ecosystem for conversational AI data governance.

## 2. Related Work

Vizgirda et al. [5] presented SocialGenPod, a prototype demonstrating decentralised generative AI

applications using Solid. SocialGenPod stores chat history, user configuration, and documents in the user's Solid Pod, decoupled from LLM providers, employing RAG over Pod-stored documents while ensuring the generative AI service receives only retrieved passages. This establishes the technical feasibility of Solid-based conversational AI but does not address the privacy failure modes we document here.

The Data Privacy Vocabulary (DPV) version 2 [6], developed by the W3C DPVCG, provides interoperable, machine-readable representations for describing personal data processing, explicitly supporting both Solid architectures and AI-related regulatory requirements including the EU AI Act. Esteves et al. [7] developed ODRL profiles for expressing consent and access control policies within Solid Pods, providing a standards-based mechanism for granular permission management.

Pandit [8] analysed Solid's relationship with GDPR compliance, identifying both opportunities and challenges including ambiguities around controllership and the legal status of Pod providers. Slabbinck et al. [9] demonstrated rule-based Web agents for enforcing usage control policies in Solid, showing how automated policy enforcement could extend beyond simple access control to govern how data is used after access is granted.

A separate body of work addresses AI agent security in centralised architectures. The OWASP Agentic Security Initiative and MAESTRO framework [10] address threats including data leakage and tool abuse in multi-agent systems, but assume centralised control and do not consider user-controlled decentralised alternatives. The gap between the Solid ecosystem's privacy capabilities and practical privacy failures in deployed conversational AI remains largely unexplored.

# 3. Use Cases

The following use cases are abstracted from documented experience with a commercial conversational AI platform over a period of several months. These are not edge cases or adversarial scenarios but arise from normal, policy-compliant operation of centralised platforms. Each follows a consistent structure: scenario, problem analysis, and regulatory implications. The use cases were formally documented and submitted to the W3C AI Knowledge Representation Community Group [2].

**Methodological note.** The use cases are derived from longitudinal autoethnographic documentation of a researcher's daily use of a commercial conversational AI platform over approximately three months (November 2025 to February 2026). Records include dated conversation transcripts, customer service correspondence, interface screenshots, and system interaction logs. Incidents were recorded contemporaneously and subsequently abstracted into generalised architectural patterns. While grounded in a single platform, the failure modes arise from structural properties common to centralised conversational AI systems and are not specific to any one provider.

## 3.1. UC-PRIVACY-001: The Assurance Gap

**Scenario.** A researcher contacts a conversational AI platform's customer service before beginning substantive use and asks: "Who will read my conversations?" The representative assures them verbally that conversations are confidential. Based on this, the researcher uses the platform extensively over months, accumulating personal health information, research data, intellectual property, and private reflections. When they later seek written confirmation of the confidentiality assurance, no corresponding policy exists. A direct inquiry to leadership receives no response.

**Problem analysis.** This reveals an *assurance gap*: the discrepancy between a user's reasonable privacy expectation -- formed through direct interaction with a company representative -- and the provider's documented obligations. Published terms reserve broad rights for data access for safety review, model improvement, and research, without communicating these at the point of interaction. No notification mechanism exists for human access to conversations.

**Regulatory implications.** Under GDPR Articles 13 and 5(1)(a), data controllers must provide transparent information about processing purposes and recipients at the time of collection [11]. The EU AI Act further requires that users be informed about how interactions are processed [12].

## 3.2. UC-AUDIT-001: Unverifiable Connector Access

**Scenario.** A user grants a conversational AI system connector access to external services (cloud storage, email). During a session, the user asks whether the agent has accessed their cloud storage. The agent reports it has not, but the user has no independent verification mechanism. The only traces are ephemeral tool-call indicators in the chat -- unsearchable, not exportable, and not independently auditable. **Problem analysis.** This use case exposes a *verification gap*. AI providers typically maintain server-side logs of connector API calls -- operational requirements such as debugging, billing, and abuse prevention strongly imply comprehensive logging of timestamps, query parameters, resources accessed, and response metadata. The user has no access to these logs and no mechanism to generate an independent audit record. This creates a structural trust asymmetry: the provider can audit everything; the user can audit nothing. Even a fully transparent agent cannot provide reliable assurance about its own data access history, because the agent's self-report is mediated by the same system whose access patterns are in question.

**Regulatory implications.** GDPR Article 15 grants data subjects the right to obtain confirmation of whether personal data is being processed [11]. The absence of user-accessible audit logs effectively prevents exercise of this right for connector-mediated access.

### 3.3. UC-SCREEN-001: Undocumented Scope Creep

**Scenario.** While working in a conversational AI interface, a user receives an unexpected browser dialog requesting permission to "see your screen" -- with no prior documentation or contextual explanation. After granting permission for research purposes, no persistent indicator confirms active sharing, no revocation mechanism is identifiable, and the AI agent itself confirms it has no information about the feature. The user's screen displayed email, professional profiles, cloud documents, and other private content.

**Problem analysis.** This documents *scope creep* -- the unannounced expansion of data access capabilities beyond the scope established at initial consent. Screen-level access represents a qualitative escalation from text-based interaction, potentially exposing the user's entire desktop environment. The absence of a persistent indicator, revocation mechanism, and documentation violates established UX norms for screen-sharing.

**Regulatory implications.** Under GDPR Article 7(2), consent must be informed and specific [11]. A generic browser permission dialog does not constitute informed consent for screen-level data capture by an AI system. Taiwan's Personal Data Protection Act similarly requires informed consent and purpose specification [13].

## 4. Solid as Privacy Architecture for Conversational AI

The Solid protocol [3, 4] provides a decentralised architecture in which users store personal data in Pods (Personal Online Data Stores) under their own control. Applications access Pod data through authenticated requests governed by access control policies (WAC or ACP). Data is represented using linked data standards, enabling interoperability across applications and providers. We assess how Solid's architecture maps to each identified privacy failure, while acknowledging limitations. Figure 1 illustrates the proposed conceptual architecture.

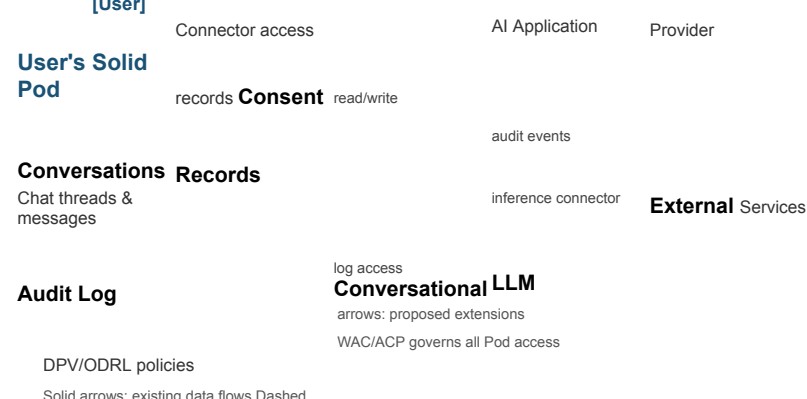

**Fig. 1.** Conceptual architecture for Solid-based conversational AI with Pod-resident conversations, audit logs, and consent records. Dashed arrows indicate proposed extensions to current Solid + AI architectures.

### 4.1. Addressing the Assurance Gap (UC-PRIVACY-001)

In a Solid-based architecture, conversation data would reside in the user's Pod rather than on provider servers. The AI application would request read/write access to specific Pod resources, governed by access control policies the user can inspect, modify, and revoke. This eliminates dependence on verbal assurances: the privacy posture is defined by machine-readable policies. SocialGenPod [5] demonstrated this for chat history, while the DPV [6] and ODRL extensions [7] provide vocabulary for expressing consent conditions and processing purposes. Slabbinck et al. [9] showed that usage control policies can be enforced by autonomous Web agents in Solid.

**Limitations.** The LLM provider necessarily receives conversation content during inference. This in-transit exposure means the provider accesses conversation content regardless of where it is stored at rest. Addressing this would require advances in privacy-preserving inference that remain experimental [5].

### 4.2. Addressing the Verification Gap (UC-AUDIT-001)

Solid suggests a natural solution: connector access events could be logged as structured resources in the user's Pod, creating a user-owned audit trail. Each entry would record timestamp, connector type, query parameters, resources accessed, and triggering conversation context. The DPV's processing records vocabulary [6] could provide the semantic framework, and MCP [14] could be extended to emit Solid-compatible audit events.

**Limitations.** A Pod-based audit log requires provider cooperation to write accurate entries. A non-cooperative provider could omit or falsify entries. Cryptographic audit mechanisms or third-party attestation would be needed -- areas for future research.

### 4.3. Addressing Scope Creep (UC-SCREEN-001)

Solid's permission model is resource-level and policy-governed: an application must request access to specific resources within granted permissions. A conversational AI wishing to access screen data would need a distinct, new permission -- one the user could evaluate against a machine-readable description per the ODRL consent profiles [7]. Changes to access scope would be auditable through the Pod's access control history.

**Limitations.** Screen sharing operates at the browser/OS level, outside Solid's protocol scope. This points to the need for complementary standards at the browser/OS level that align with Solid's permission model.

## 5. Discussion

| Use Case | Privacy Failure | Solid Mechanism | Open Gap |
|---|---|---|---|
| UC-PRIVACY-00 1 | Assurance gap | Pod storage, WAC/ACP, DPV consent records | In-transit exposure during inference |
| UC-AUDIT-001 | Verification gap | Pod-based audit logs, DPV processing records | Provider cooperation, log integrity |
| UC-SCREEN-001 | Scope creep | Resource-level permissions, ODRL consent profiles | Browser/OS-level capture outside Solid scope |

**Table 1.** Summary mapping of use cases to Solid mechanisms and open gaps.

Table 1 summarises the mapping between use cases, Solid mechanisms, and remaining gaps. The use cases collectively illustrate that conversational AI data is qualitatively different from personal data typically discussed in Solid contexts. Conversational data is *involuntarily comprehensive* -- it accumulates not because the user deliberately enters it into a form, but because sustained natural language interaction inevitably discloses personal information. This makes conversational AI a particularly strong candidate for

user-controlled data governance.

Solid's existing mechanisms address *storage and access control* effectively but do not address *in-transit data exposure* during AI inference. The LLM must receive the user's input to produce a response. In the interim, Solid can ensure that conversations are stored under user control and that the provider's post-inference retention and use of conversation data is governed by enforceable policies.

The regulatory landscape increasingly demands the transparency and user control that Solid provides. The EU AI Act [12] requires providers to document data governance practices and enable user rights exercise. The Data Governance Act [15] establishes frameworks for personal data spaces that align with Solid's Pod architecture. Pandit [8] and the DPV [6] have laid groundwork for GDPR-compliant Solid implementations; extending this to conversational AI is a natural next step.

Finally, the emergence of agent protocols such as MCP [14] creates an opportunity to embed Solid-compatible privacy controls at the protocol level. If MCP servers emitted structured audit events to the user's Pod, the verification gap could be addressed architecturally rather than relying on provider goodwill.

## 6. Proposed Research Agenda

Based on the use cases and analysis presented, we propose the following directions for further exploration within the Solid community:

**(1) Pod-based conversation storage.** Extend SocialGenPod [5] to support full conversation lifecycle management -- creation, retrieval, search, export, and deletion of conversation threads as Solid resources -- and evaluate performance implications for real-time interactions.

**(2) Standardised audit logging for AI connectors.** Define a Solid-compatible vocabulary for AI connector access audit logs, building on the DPV's processing records [6], with integration pathways for MCP [14].

**(3) Machine-readable consent records for AI interactions.** Apply the DPV's consent modelling and ISO/IEC 27560 [16] to conversational AI, enabling granular consent for conversation storage, model training, human review, and connector access.

**(4) Scope change notification and re-consent.** Develop mechanisms within Solid's access control framework for detecting and notifying users when an application's data access scope changes. **(5) Cross-jurisdictional privacy policy expression.** Leverage the DPV's multi-jurisdictional legal extensions [6] to enable users to express privacy requirements under their applicable law and have these enforced through Solid access control policies.

## 7. Conclusion

The three use cases presented -- the assurance gap, the verification gap, and scope creep -- arise from structural properties of centralised conversational AI platforms. Solid's decentralised architecture offers a principled foundation for addressing these challenges through user-controlled Pods, granular access control, and linked data standards. Prior work, notably SocialGenPod [5] and the DPV [6], demonstrates feasibility, though significant gaps remain around in-transit exposure, audit log integrity, and browser-level capture. We offer these use cases and the associated research agenda as an invitation to the Solid community to engage with the privacy demands of conversational AI -- a domain where user-controlled data governance is both urgent and well-suited to Solid's architectural strengths.

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
