# OpenReview forum: "Privacy Challenges in Conversational AI: Three Use Cases and Prospects for Decentralised Data Governance with Solid"
_SolidProject.org/SoSy/2026/Privacy_Session — SoSy2026-Privacy Paper_

### Official Review · ~Andres_Chomczyk_Penedo1 · 2026-02-25
**Good and suitable paper for the conference**

**Rating:** 8
**Confidence:** 5

**Review:**

-	Quality:
o	The paper is a strong submission for the workshop, with concrete empirical work that triggers active discussion.
-	Clarity:
o	The paper is quite clear on (i) its objectives, (ii) its methodology, and (ii) its intended results/outcome, making it scientifically sound.
o	The only missing element is the elaboration of a clear research question to articulate the well-polished elements mentioned above. This, by itself, does not invalidate the research conducted nor the quality of the paper but could be improved in the final version of the paper.
o	Language used is adequate for the expected level of the submissions.
o	Nevertheless, it would be beneficial to conduct further proofreading to correct some minor typos, such as missing spaces, and some wording used.
-	Originality:
o	This reviewer is not aware of any other paper tackling the problem identified by the author(s). As such, it constitutes a clear novel work that explores an uncharted area.
-	Significance:
o	The paper clearly lays out a concrete future research agenda, which this reviewer is in accordance with.
o	As a suggestion, the development of a methodology for conducting this type of research could be another relevant path to explore in the future, particularly if the author(s) intend to expand their exploration.

---

### Official Review · ~Harshvardhan_J._Pandit1 · 2026-03-05
**discussion-oriented work on Solid with use-cases and a fair grounding on considerations**

**Rating:** 7
**Confidence:** 5

**Review:**

Summary: The work describes three use-cases where users interact with an external service involving AI, and through these surface three privacy issues. The work then discusses the implications of these when using Solid, with a mapping to mechanisms and the gaps.

The work is coherent, clearly useful for discussions, and contains sufficient aspects to be of interest to the audience. Below are suggestions for improvements.

The categories of issue provided (assurance, verification, scope creep) need independent definitions. Otherwise it is unclear on what basis do they apply or can be explored. E.g. assurance of information and verification of information are close concepts and related, and instinctively it could be assumed what they mean but this should be explicitly defined. Further, it is unclear why only three cases exist, or why these are chosen and are good examples. A good method here would be to take normative requirements (e.g. GDPR, AI Act, or an ISO privacy framework) and then to derive the issues and examples based on what is required or missing from implementations.

In Section 4, the Fig.1 is missing arrows, so it is unclear what is being depicted. I have tried my best to consider the relations based on discussions. In this, the use of "Consent" is potentially misleading -- Solid has "Permissions" which the user controls (ACL, etc.) and since the rest of the discussions are based in GDPR, it would imply here that this consent is the same as GDPR's consent, which is neither required by the platform nor supported by it (e.g. notice, withdrawal are missing).

Further to above, the identified 'gaps' and the mapping to existing mechanisms as well as additional needs are currently argument-based. It is always better to ground privacy analysis in existing work, especially in standards (e.g. ISO 27000 series) or in law (e.g. GDPR guidelines) so that it is clear which aspects are legally relevant and which are not. This is important to recognise 1) the relevant actors whose responsibilities it is to implement these; and 2) the limitations of the law to recognise the gaps and address them. This can then help ground the proposed research agenda in Section 6, e.g. rather than making specific statements about SocialGenPod and only for conversation, to talk about implementing PIMS standards as a way to let users have governance-level control over their own data. Or replacing the use of 27560 in (3) as a record in (2), and using 29184 as notice in (3) which also addresses the issue of what is declared to the user (whether manual or machine-readable). Both of these are then useful in (5) where scope change can be detected based on sufficiently changed requests or policies.

Note: 3.2 rather stretches the requirement of GDPR Article 15 which only states that the data subject should have 1) confirmation of processing; and 2) copy of the processed data. Stating that this means the user must have access to the logs itself is conditional upon the provider not having other means to give this information.

- Note that Solid isn't a W3C standard, and neither is the DPV (both are CG outputs)
- 3.1 The article for GDPR is provided, but not for AI Act

---

### Decision · Program_Chairs · 2026-03-09

Accept (Paper)